# Spatial Analysis of Mountain and Lowland Anoa Habitat Potential Using the Maximum Entropy and Random Forest Algorithm

**Diah Ardiani** [1], **Lalu Muhamad Jaelani** [1,*], **Septianto Aldiansyah** [2], **Mangapul Parlindungan Tambunan** [2], **Mochamad Indrawan** [3] and **Andri A. Wibowo** [3]

1 Department of Geomatics Engineering, Institut Teknologi Sepuluh Nopember, Keputih, Sukolilo, Surabaya 60111, Indonesia; diahardiani.18033@mhs.its.ac.id
2 Department of Geography, Faculty of Mathematics and Natural Sciences (FMIPA), Universitas Indonesia, Depok 16424, Indonesia; septianto.aldiansyah@ui.ac.id (S.A.); mangapul.parlindungan@ui.ac.id (M.P.T.)
3 Department of Biology, Faculty of Mathematics and Natural Sciences (FMIPA), Universitas Indonesia, Depok 16424, Indonesia; mochamad.indrawan@ui.ac.id (M.I.); wibowohealth@hotmail.com (A.A.W.)
* Correspondence: lmjaelani@geodesy.its.ac.id

**Abstract:** The Anoa is a wild animal endemic to Sulawesi that looks like a small cow. Anoa are categorized as vulnerable to extinction on the IUCN red list. There are two species of Anoa, namely Lowland Anoa (*Bubalus depressicornis*) and Mountain Anoa (*Bubalus quarlesi*). In this study, a comparison of potential habitat models for Anoa species was conducted using Machine Learning algorithms with the Maximum Entropy (MaxEnt) and Random Forest (RF) methods. This modeling uses eight environmental variables. Where based on the results of *Bubalus quarlesi* potential habitat modeling, the RF 75:25 model is the best algorithm with the highest variable contribution, namely humidity of 82.444% and a potential area of 5% of Sulawesi Island, with an Area Under Curve (AUC) of 0.987. Meanwhile, the best *Bubalus depressicornis* habitat potential model is the RF 70:30 algorithm, with the highest variable contribution, namely population of 88.891% and potential area of 36% of Sulawesi Island, with AUC 0.967. This indicates that Anoa extinction is very sensitive to the presence of humidity and human population levels.

**Keywords:** Anoa; habitat potential; MaxEnt; RF

## 1. Introduction

Anoa (*Bubalus* sp) is a wild animal endemic to Sulawesi that is similar to cattle or buffalo but with a smaller size [1]. The government had protected the Anoa since before the country's independence when the colonial government included the Anoa as a protected animal. In the International Union for Conservation of Nature (IUCN) red list, the Anoa is classified as endangered and is included in Appendix A of CITES [2]. Anoa are highly valued on a regional, national and international scale because they are rare, endemic, vulnerable to extinction and have a unique and complex evolutionary history.

Opinions on the Anoa species vary, with some arguing that the Anoa species was related to the bull by a wildlife expert named Groves in 1969. Taxonomists also dispute the number of Anoa species in Sulawesi. Some claim that there are two Anoa species, namely Lowland Anoa (*Bubalus depressicornis*) and Mountain Anoa (*Bubalus quarlesi*) [3,4]. Others argue that only one species of Anoa has two or three subspecies: *Bubalus depressicornis*, *B.d. quarlesi* and *B.d. fergusoni* [5].

The difference in opinion between the number of Anoa species can be analyzed by looking at the suitable habitat conditions between Mountain Anoa and Lowland Anoa. Previous research by Jaelani [6,7] showed that by using one Anoa species (without differentiating between them). With the help of increasingly sophisticated technology, it is possible

to analyze the differences in habitat types between Mountain Anoa and Lowland Anoa. This study was performed by modeling the habitat of the two types of Anoa according to existing environmental variables. Habitat modeling of Mountain Anoa and Lowland Anoa was conducted using a comparison of the Maximum Entropy (MaxEnt) and Random Forest (RF) models. In recent years, several studies have compared the use of RF and MaxEnt models in mapping amphibian distribution in China by Zhao [8], showing that the RF model is slightly better than the MaxEnt model.

Each model has advantages to be used as a species distribution modeling method. RF modeling is effective in forming models with limited training data samples, and it is not sensitive to training data containing outliers [9]. It also can minimize over-fitting [10]. If RF can handle species' absence and presence, MaxEnt modeling can use only species presence data and handle irregular environmental data. The MaxEnt probability distribution has a concise mathematical definition, making it easy to analyze [11].

This study will compare the spatial modeling of Mountain Anoa and Lowland Anoa distribution using Machine Learning of MaxEnt and RF algorithms. In addition to comparing the animal distribution models, this study also compares the comparison ratio between the training and the testing dataset according to previous research conducted by [12]. This study will obtain the best model from the algorithm used, the training dataset ratio and testing dataset comparisons.

The output of each model is habitat suitability, and it is hoped that this study can be used for conservation efforts for Anoa according to their species and habitat so that they can breed well. Biodiversity management is in line with Law No. 5 of 1994, which is directed at Indonesia's commitment to implement the three main objectives of the Convention on Biological Diversity, including the conservation and sustainable utilization of biodiversity components [13].

## 2. Materials and Methods

This study was performed on Sulawesi Island, the world's eleventh-largest island with an area of 180,680.7 km$^2$ (01.7° N–05.8° S and 112.7° E–125.3° E) [14]. The island is situated north of the Lesser Sunda Islands, south of Mindanao, west of the Maluku Islands and east of Borneo. This area has a maximum altitude of 3478 m above sea level, and is administratively part of six provinces: North Sulawesi, South Sulawesi, West Sulawesi, Central Sulawesi, Southeast Sulawesi and Gorontalo [14], as presented in Figure 1.

Anoa have specific paths or corridors in the forest that can connect one type of habitat to another or connect one resource with other resources Anoa needs, such as food, drink, wallowing, rest and shelter. Anoa's paths and movements can be easily identified from the footprints and dirt on the trails [1]. The data required for this study included in-situ presence data on the coordinates of Anoa tracks in Appendix A (Table A1). In situ data on Anoa presence was obtained from several related journal sources as well as field research by the team in 2021. The environmental variables are described in Table 1. SRTM DEM 30-m data was obtained from the official USGS website https://earthexplorer.usgs.gov/ (accessed on 5 September 2022) [15]. Land Surface Temperature (LST) and Normalize Difference Vegetation Index (NDVI) data from MOD11A1 product. Land cover data from the ESA 10 m world cover product. The European Space Agency (ESA) product of world cover 10 m 2020 provides global land cover maps for 2020 at 10 m resolution based on Sentinel-1 and Sentinel-2 data. Air humidity data was retrieved from FLDAS 11 km, human population data from WorldPop 2020, and 250,000 scale road and water vector data obtained from the Indonesia Geospatial Information Agency (BIG) official website https://tanahair.indonesia.go.id/portal-web (accessed on 1 November 2021).

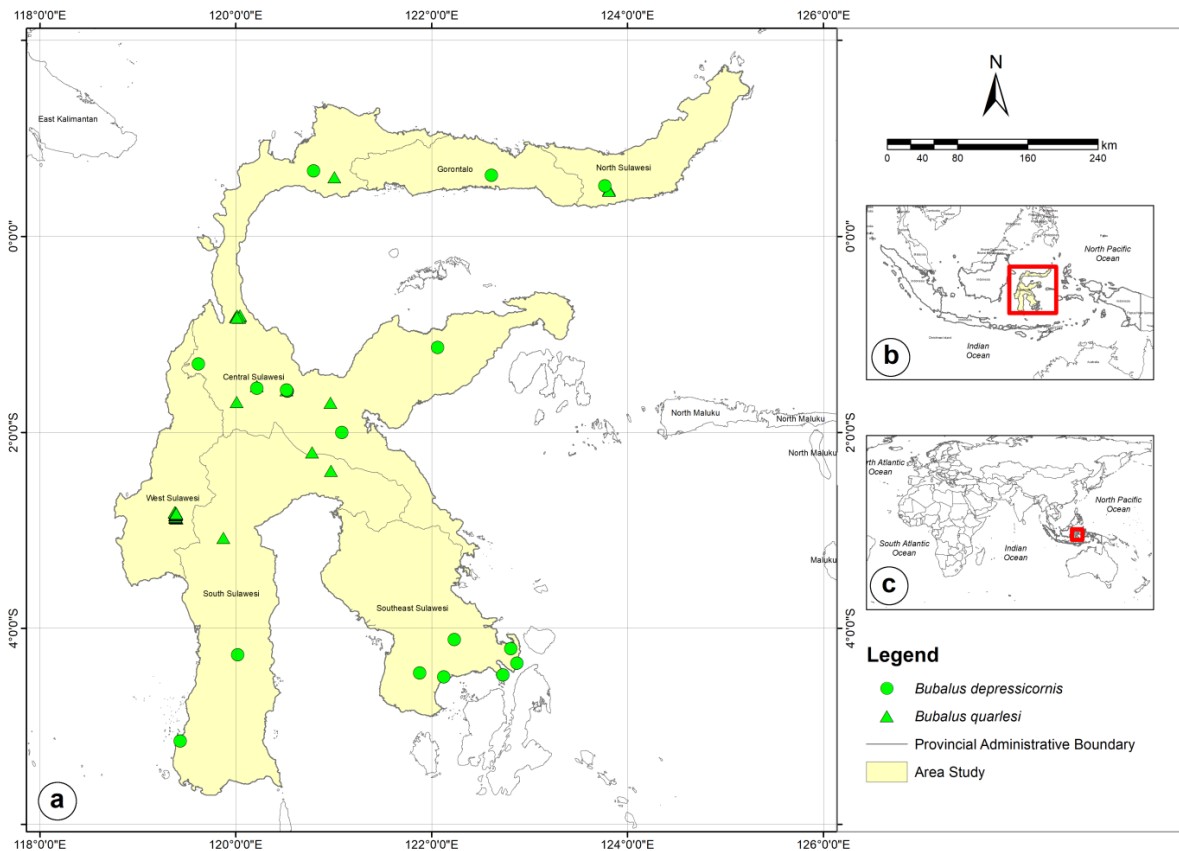

**Figure 1.** Research area: (**a**) Sulawesi Island, (**b**) location of Sulawesi Island on Indonesia Map, (**c**) location of Indonesia on the World Map.

**Table 1.** Environmental variables used in the MaxEnt and RF models.

| No | Variables | Cell Size (m) | Source | Class | Range Data |
|----|-----------|---------------|--------|-------|------------|
| 1 | DEM/Elevation | 30 | Shuttle Radar Topography Mission (SRTM) | 1 | 0–1000 m |
| | | | | 2 | 1000–1500 m |
| | | | | 3 | 1500–2000 m |
| | | | | 4 | 2000–2500 m |
| | | | | 5 | >2500 m |
| 2 | Temperature | 100 (resampled to 30) | MOD11A1 Version 6 product | 1 | <20 °C |
| | | | | 2 | 20–25 °C |
| | | | | 3 | 25–30 °C |
| | | | | 4 | >30 °C |
| 3 | Vegetation Index | 100 (resampled to 30) | MOD11A1 Version 6 product | 1 | <0 |
| | | | | 2 | 0–0.25 |
| | | | | 3 | 0.25–0.50 |
| | | | | 4 | 0.50–0.75 |
| | | | | 5 | 0.75–1 |
| 4 | Land Cover | 10 (resampled to 30) | ESA WorldCover | 10 | Trees |
| | | | | 20 | Shrubland |
| | | | | 30 | Grassland |
| | | | | 40 | Cropland |
| | | | | 50 | Built-up |
| | | | | 60 | Barren/Sparse Vegetation |
| | | | | 80 | Open Water |
| | | | | 90 | Herbaceous Wetland |
| | | | | 95 | Mangroves |

**Table 1.** *Cont.*

| No | Variables | Cell Size (m) | Source | Class | Range Data |
|----|-----------|---------------|--------|-------|------------|
| 5 | Transportation | 30 | BIG | 1 | 0–500 m |
| | | | | 2 | 501–1000 m |
| | | | | 3 | 1001–1500 m |
| | | | | 4 | 1501–2000 m |
| | | | | 5 | >2000 m |
| 6 | Water | 30 | BIG | 1 | 0–500 m |
| | | | | 2 | 501–1000 m |
| | | | | 3 | 1001–1500 m |
| | | | | 4 | 1501–2000 m |
| | | | | 5 | >2000 m |
| 7 | Human Population | 100 (resampled to 30) | WorldPop | | 0–1553 people/pixel |
| 8 | Relative Humidity | 11,000 (resampled to 30) | FLDAS | | 9.45–19.34% |

The software used for data processing in this study is Google Earth Engine to process environmental variables, MaxEnt version 3.4.1(Steven J. Phillips; New York, USA) [16] and R version 4.1.1 [17] with the packages "rgdal" for spatial data processing, "raster" for raster processing, "RStoolbox" for image analysis and plotting spatial data, "caret" for machine learning and "e1071" for RF process. The last is ArcMap 10.8 to visualize maps.

The data processing stages in this study are depicted in the flow chart in Figure 2. There is a preparation stage to download the dependent variable data in the form of Anoa distribution coordinates, which will be stored in *.csv and *.shp formats.

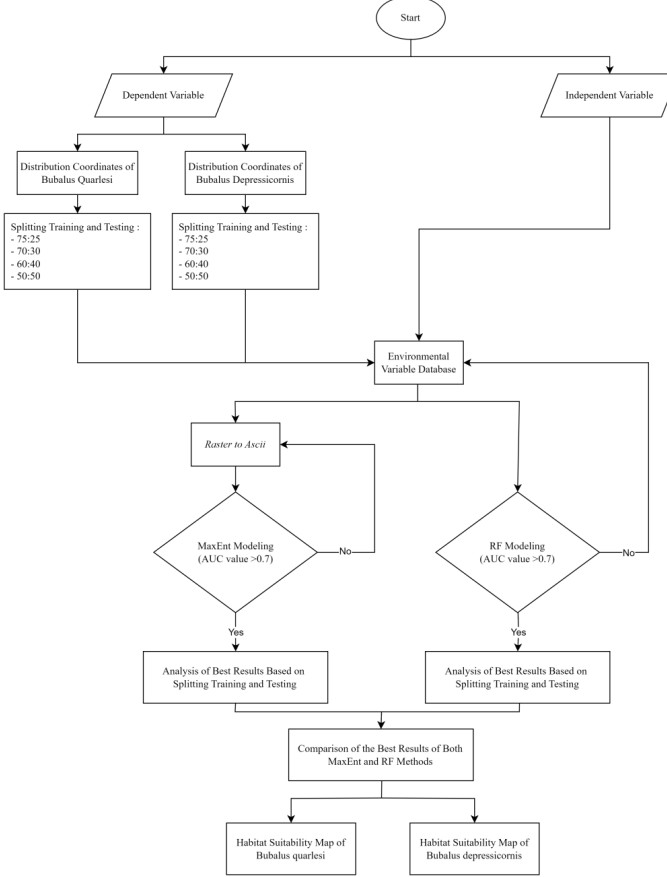

**Figure 2.** Processing flow-chart.

The next stage is processing the SRTM DEM using Google Earth Engine [18] into elevation information in *.tif format with a pixel size of 30 m. Reclassify was performed for categorical altitude data division [7], i.e., class 1 (0–1000 m), 2 (1000–1500 m), 3 (1500–2000 m), 4 (2000–2500 m) and 5 (>2500 m). Continued processing of Terra-MODIS image data was processed using Google Earth Engine Sulawesi Island into temperature information (LST) with *.tif format. Reclassify was performed for categorical LST data division [7], i.e., class 1 (<20 °C), 2 (20–25 °C), 3 (25–30 °C) and 4 (>30 °C). Terra-MODIS image data processing was also processed using Google Earth Engine on the island of Sulawesi to produce a vegetation index (NDVI) in *.tif format. Categorical NDVI data division was divided into five classes [7], i.e., class 1 (<0), 2 (0–0.25), 3 (0.25–0.5), 4 (0.5–0.75) and 5 (0.75–1). The next stage was processing ESA world cover data using Google Earth Engine Sulawesi Island into land cover information in *.tif format. Reclassify was performed for categorical land cover data division, i.e., the classes of trees, shrubs, grasslands, agricultural land, developed land, barren vegetation, waters, herbaceous wetlands and mangroves. Furthermore, vector data of roads and waters was processed using ArcMap to produce euclidean distance in *.tif format. Vector data processing was then reclassified for categorical data division [7], i.e., class 1 (0–500 m), 2 (501–1000 m), 3 (1001–1500 m), 4 (1501–2000 m) and 5 (>2000 m). Famine Early Warning Systems Network (FEWS NET) Land Data Assimilation System (FLDAS) data in 2021 was processed using Google Earth Engine and then averaged for one year. Because the spatial resolution of FLDAS data is 11 km, an interpolation was carried out with environment settings related to the resulting cell size of 30. Finally, the WorldPop data for one country was subsetted according to the study area used to obtain human population data.

The numerical data of the analysis variables extracted from Anoa presence points can be correlated with each variable. Based on Table 2, the highest positive correlation value is the correlation between humidity and temperature with a correlation value of 0.932.

**Table 2.** Correlation between variables.

| | Elevation | Temperature | Vegetation | Relative Humidity | Human Population | Land Cover | Water | Transportation |
|---|---|---|---|---|---|---|---|---|
| Elevation | 1 | | | | | | | |
| Temperature | −0.851 | 1 | | | | | | |
| Vegetation | −0.672 | 0.478 | 1 | | | | | |
| Relative Humidity | −0.912 | 0.932 | 0.513 | 1 | | | | |
| Human Population | −0.159 | 0.345 | −0.255 | 0.218 | 1 | | | |
| Land Cover | −0.237 | 0.403 | 0.038 | 0.353 | 0.460 | 1 | | |
| Water | −0.501 | 0.482 | 0.379 | 0.541 | 0.051 | −0.043 | 1 | |
| Transportation | 0.467 | −0.666 | −0.260 | −0.549 | −0.367 | −0.272 | −0.316 | 1 |

Since MaxEnt processing used ASCII format (*.asc) data, all environmental raster data (*.tif) was then converted to it, whereas the Anoa distribution coordinates needed to be stored in CSV format (*.csv). All variables were recorded at a pixel size of 0.00026949459° by 0.00026949459°, equivalent to 30 m by 30 m. MaxEnt modeling can identify wildlife distribution and habitat selection by considering the location of occurrence [19]. MaxEnt generates a map that shows the likelihood of the studied species being found in a particular area. MaxEnt calculations produce habitat suitability indicated by a range of values between 0 and 1; the closer to 1, the more suitable the habitat for the animals studied [20]. There are four class divisions for habitat suitability modeling based on Kumar [21].

The RF processing used variables in the format (*.tif) and Anoa distribution coordinates in the format (*.shp). The variable map is presented in Figure 3. Then, the potential habitat model results for Mountain Anoa and Lowland Anoa were analyzed based on the best algorithm from the AUC value that is closer to 1 [6].

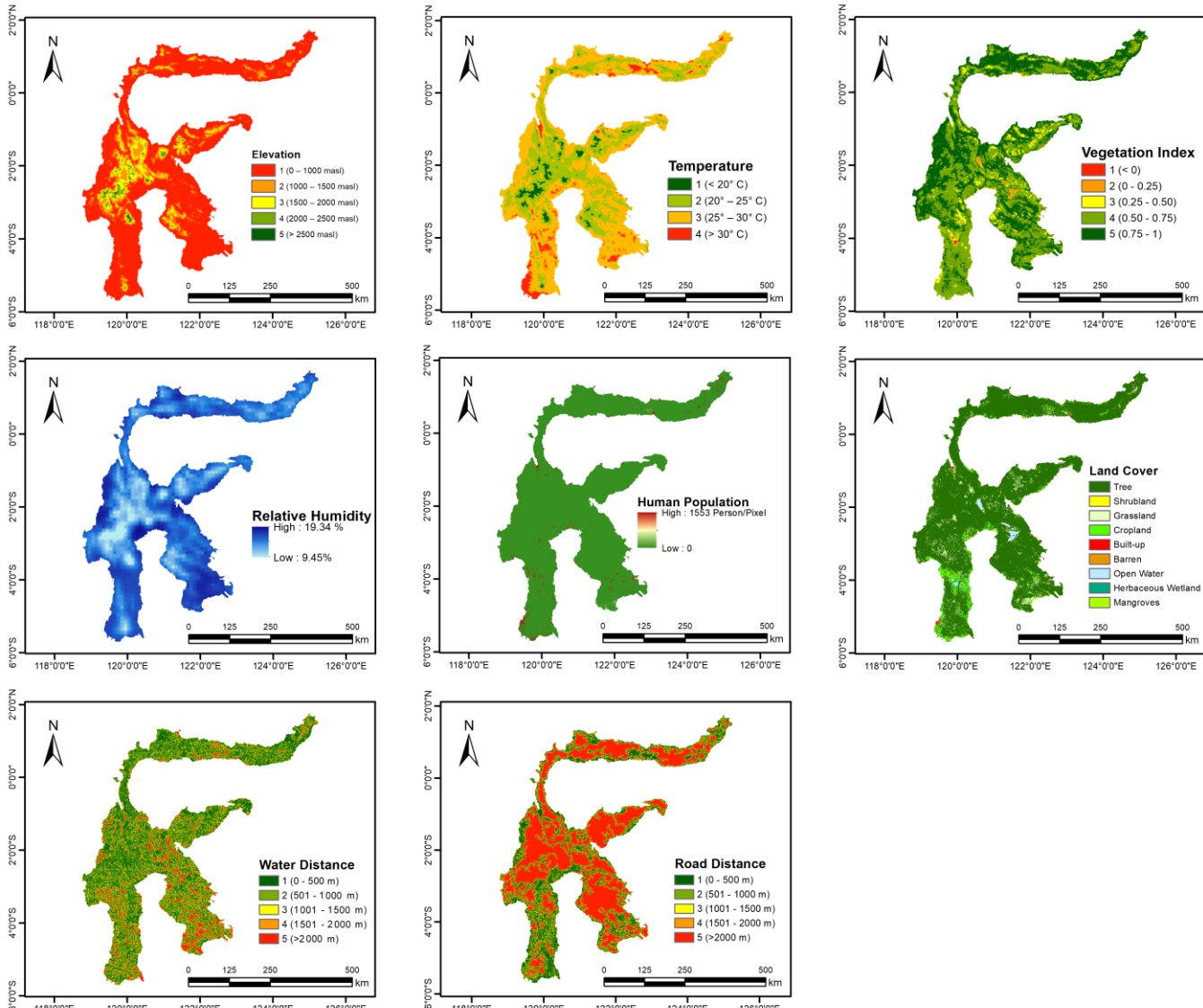

**Figure 3.** Environmental variables for predicting the potential of Anoa habitat.

The accuracy assessment for each model variable was measured by the Area Under Curve (AUC) of the Receiver Operating Characteristics (ROC) curve. The minimum acceptable model accuracy and performance standard is when the Random Prediction value reaches above or is equal to 0.5 (AUC = 0.5) [22]. The AUC value ranges from 0 to 1. When the AUC value is less than 0.7, the prediction accuracy of the model is usually considered average; when the AUC value is between 0.7 and 0.9, the prediction accuracy of the model is high; and when the AUC value is greater than or equal to 0.9, the prediction accuracy of the model is very good [23,24]. Sensitivity can be used as a measure of the proportion of "true positives" that are correctly identified. Meanwhile, specificity is defined as a measure of the proportion of "true negatives" that are correctly identified [25].

In RF statistics, there are additional accuracy results obtained from Kappa and Detection Rate. Where Kappa is a measure that states the consistency of measurements made by two raters or the value of consistency between two measurement methods. In the evaluation of the ML model, what is meant by raters here is prediction and observation. This parameter is generally used for two-class classification. Using the kappa confusion matrix can be determined by equation 1 [26]. While the detection rate is the number of

declared animals that have been found compared to the number of animals that are still estimated in a certain area [27].

$$K = \frac{2 \cdot (TP \cdot TN - FP \cdot FN)}{(TP + FP) \cdot (FP + TN) + (TP + FN) \cdot (FN + TN)} \tag{1}$$

## 3. Results

### 3.1. MaxEnt Modeling Results

As shown in Figure 4. Experiments were carried out to get the best results by comparing the splitting ratio of training and testing datasets, i.e., 75:25, 70:30, 60:40 and 50:50.

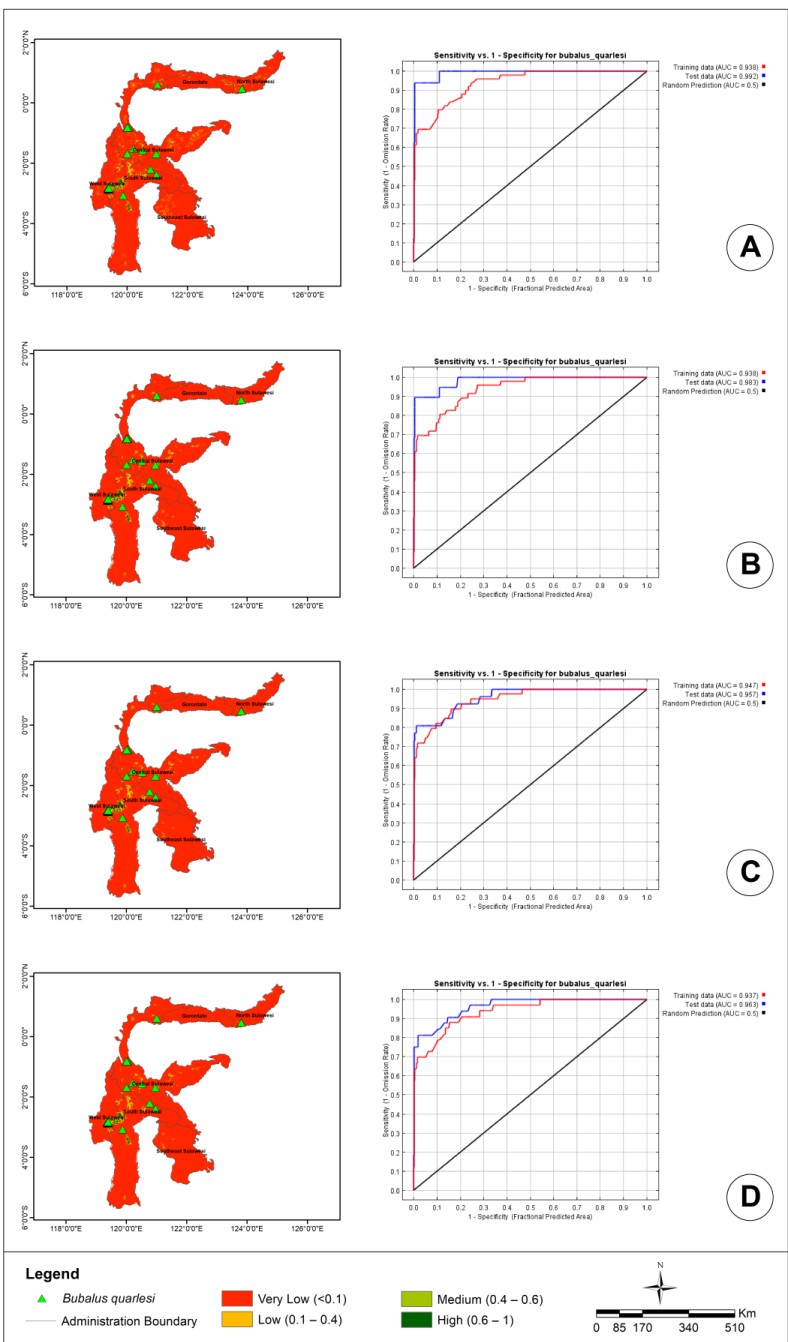

**Figure 4.** Map (left) and ROC/AUC curve (right) of the *Bubalus quarlesi* habitat potential model with MaxEnt algorithm (training-test ratio in %, (**A**) 75:25, (**B**) 70:30, (**C**) 60:40, (**D**) 50:50).

In the case of this MaxEnt model, the best AUC result for *Bubalus quarlesi* is shown as 0.947 in Figure 4 with a training and testing ratio of 60:40. This value shows good performance as it is quite close to the highest value of 1, which indicates that the MaxEnt model at a training-testing ratio of 60:40 has a reasonably good ability to distinguish the potential habitat classes of Mountain Anoa (*Bubalus quarlesi*).

In the case of this MaxEnt model, the best AUC result for *Bubalus depressicornis* was shown as 0.824 in Figure 5 with a training and testing ratio of 60:40. This value shows good performance as it is quite close to the highest value of 1, which indicates that the MaxEnt model at a training-testing ratio of 60:40 has a reasonably good ability to distinguish the potential habitat classes of Lowland Anoa (*Bubalus depressicornis*).

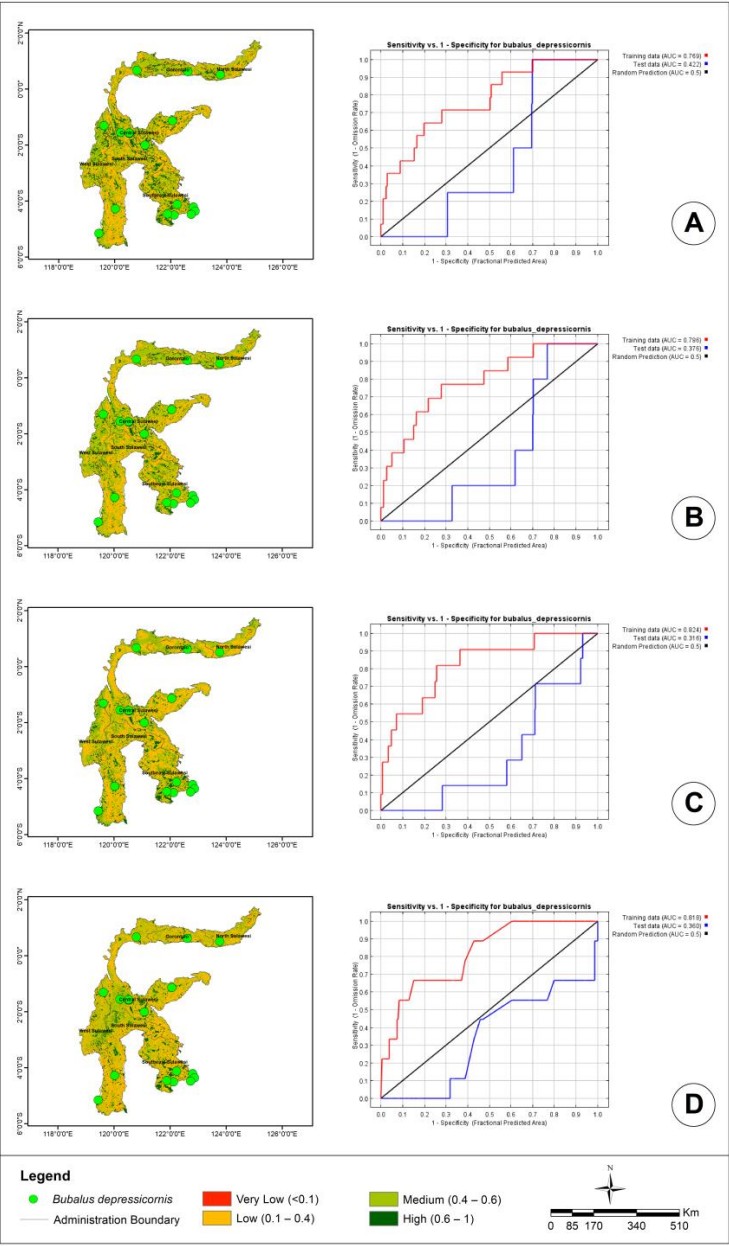

**Figure 5.** Map and ROC/AUC curve of the *Bubalus depressicornis* habitat potential model with MaxEnt algorithm (training-test ratio in %, (**A**) 75:25, (**B**) 70:30, (**C**) 60:40, (**D**) 50:50).

### 3.2. RF Modeling Results

There are four class divisions for habitat suitability modeling as shown in Figure 6.

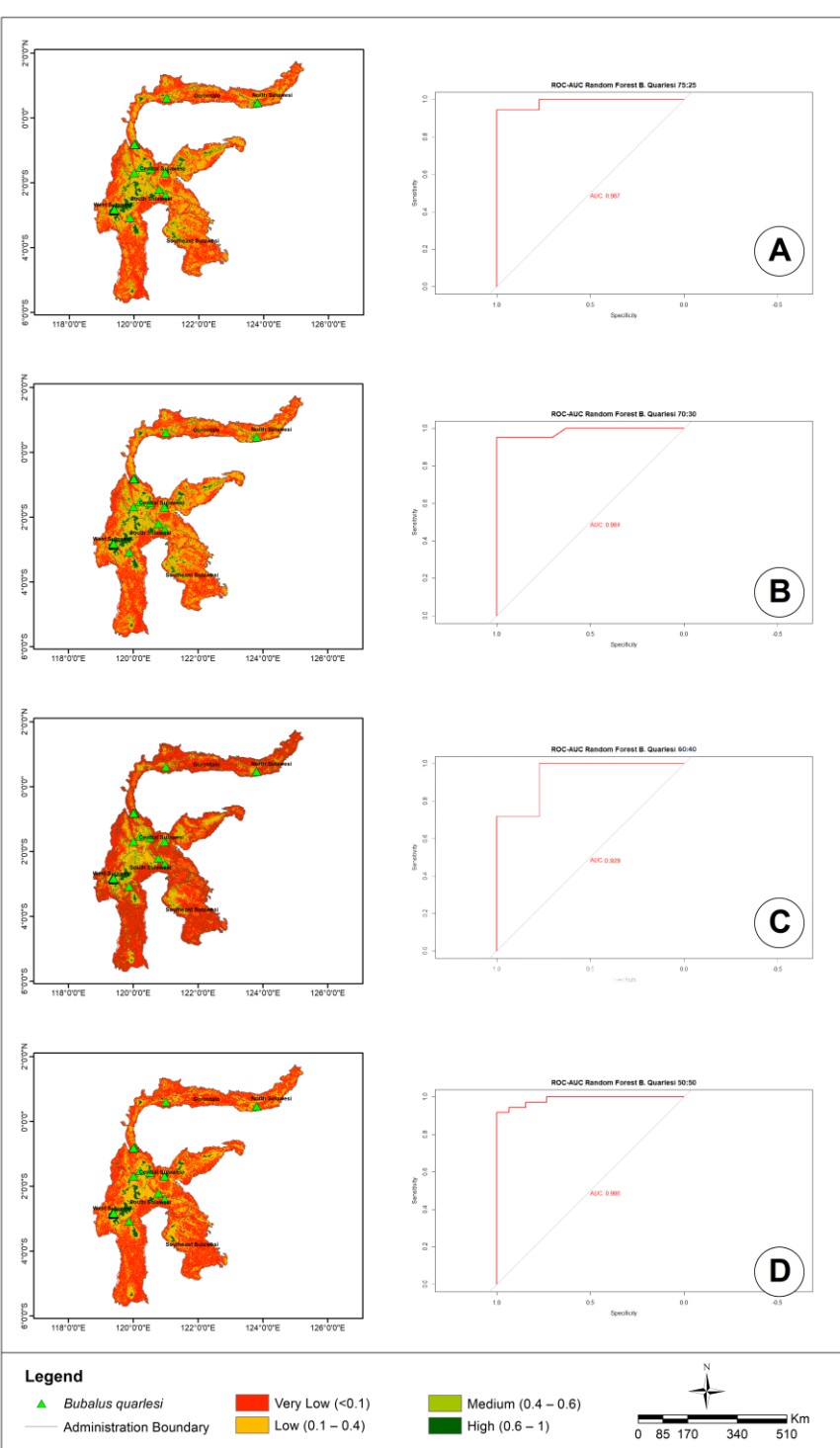

**Figure 6.** Map and ROC/AUC curve of the *Bubalus quarlesi* habitat potential model with RF algorithm (training-test ratio in %, (**A**) 75:25, (**B**) 70:30, (**C**) 60:40, (**D**) 50:50).

In the case of this RF model, the best AUC result for *Bubalus quarlesi* was shown as 0.987 in Figure 6 with a training and testing ratio of 75:25. This value shows good performance as it is quite close to the highest value of 1, which indicates that the RF model at a training-testing ratio of 75:25 has a reasonably good ability to distinguish the potential habitat classes of Mountain Anoa (*Bubalus quarlesi*).

In the case of this RF model, the best AUC result for *Bubalus depressicornis* is shown as 0.967 in Figure 7 with a training and testing ratio of 70:30. This value shows good

performance as it is quite close to the highest value of 1, which indicates that the RF model at a training-testing ratio of 70:30 has a reasonably good ability to distinguish the potential habitat classes of Lowland Anoa (*Bubalus depressicornis*).

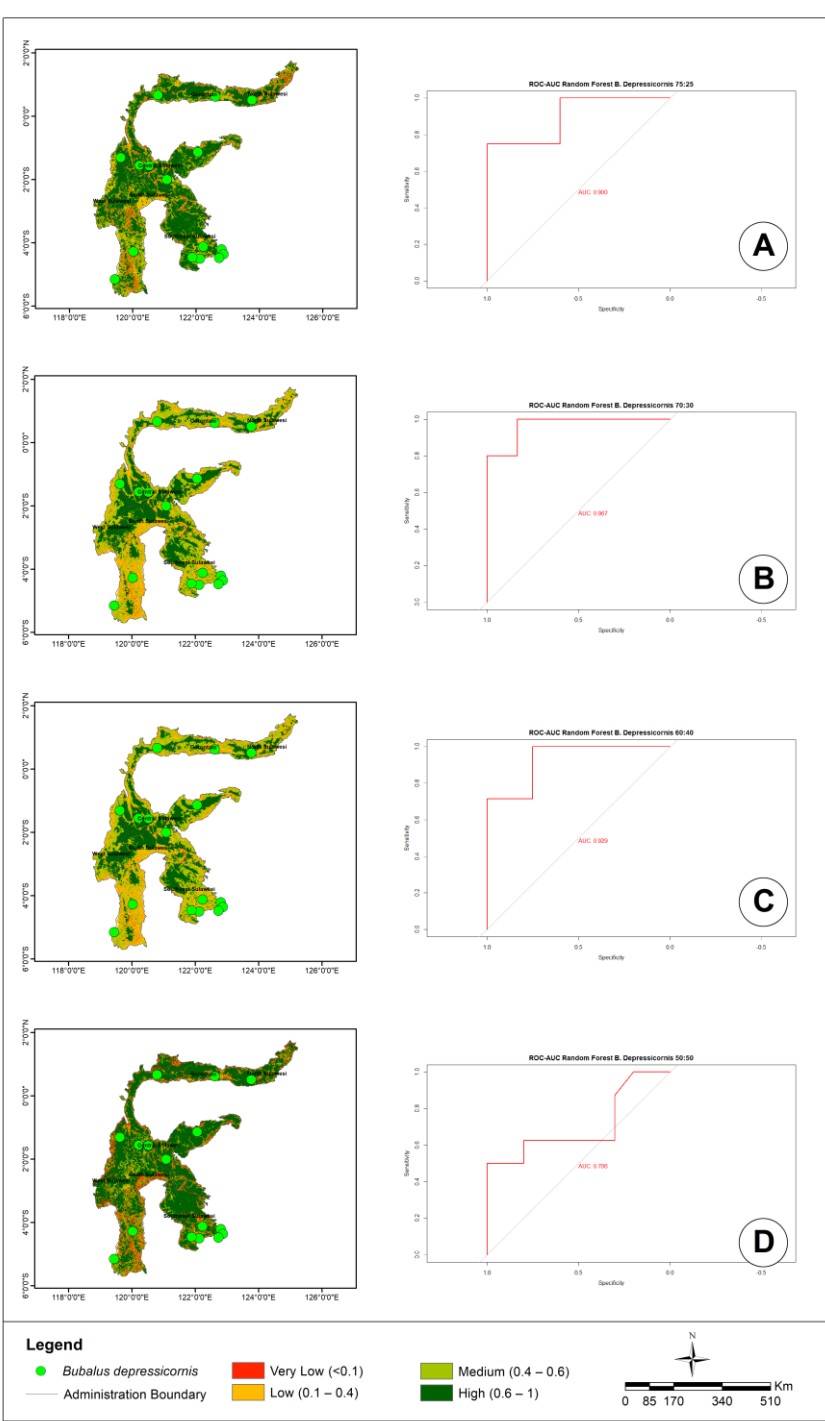

**Figure 7.** Map and ROC/AUC curve of the *Bubalus depressicornis* habitat potential model with RF algorithm (training-test ratio in %, (**A**) 75:25, (**B**) 70:30, (**C**) 60:40, (**D**) 50:50).

*3.3. Best Results*

Based on the modeling results, the RF model with a training and testing ratio of 75:25 was the best algorithm for the Mountain Anoa habitat potential model with a high potential habitat area of 837,400 ha (5% of the island of Sulawesi) in Table 3 showing the results of the

area of potential habitat according to its classification in each province. While for the AUC value of 0.987; Accuracy 0.975; Kappa 0.992; Sensitivity 0.944; Specificity 1; and Detection Rate 0.425.

**Table 3.** Potential *Bubalus quarlesi* habitat area in each province on Sulawesi Island (in hectares).

| Province / Class | Gorontalo | West Sulawesi | South Sulawesi | Central Sulawesi | Southeast Sulawesi | North Sulawesi |
|---|---|---|---|---|---|---|
| Very Low | 676,577 | 784,321 | 2,247,574 | 2,283,917 | 1,337,869 | 666,963 |
| Low | 473,689 | 624,690 | 1,602,293 | 2,719,151 | 1,164,064 | 496,856 |
| Medium | 17,705 | 93,083 | 227,439 | 329,480 | 75,105 | 25,420 |
| High | 20,383 | 129,326 | 285,650 | 295,184 | 71,256 | 35,613 |

Meanwhile, the best model for Lowland Anoa habitat potential was the RF algorithm with a training and testing ratio of 70:30 with a high potential area of 6,015,700 ha (36% of the island of Sulawesi). Table 4 shows the results of the area of potential habitat according to its classification in each province. While for the AUC value of 0.967; Accuracy 0.909; Kappa 0.814; Sensitivity 0.8; Specificity 1; and Detection Rate 0.364.

**Table 4.** Potential *Bubalus depressicornis* habitat area in each province on Sulawesi Island (in hectares).

| Province / Class | Gorontalo | West Sulawesi | South Sulawesi | Central Sulawesi | Southeast Sulawesi | North Sulawesi |
|---|---|---|---|---|---|---|
| Very Low | 23,773 | 17,105 | 179,476 | 91,704 | 23,915 | 33,029 |
| Low | 317,990 | 372,482 | 1,638,132 | 1,238,548 | 638,400 | 347,087 |
| Medium | 504,452 | 584,004 | 1,185,045 | 1,865,110 | 1,144,664 | 465,104 |
| High | 342,139 | 657,829 | 1,360,304 | 2,432,371 | 841,315 | 379,632 |

The percentage of environmental variables that support the modeling of potential habitat for Mountain Anoa (*Bubalus quarlesi*) and Lowland Anoa (*Bubalus depressicornis*) is shown in Figure 8.

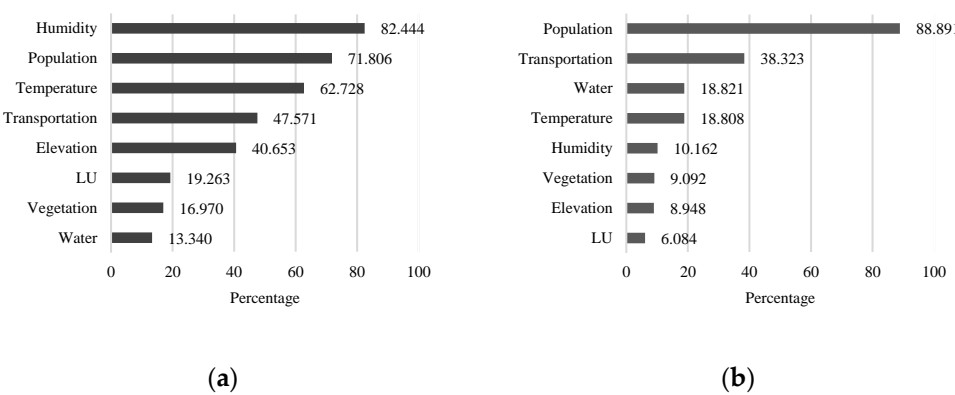

**Figure 8.** Contribution Variables in RF Modeling (**a**) *Bubalus quarlesi* 75:25, (**b**) *Bubalus depressicornis* 70:30.

Based on the graph in Figure 8a, the parameter or variable with the highest relative contribution to the Mountain Anoa habitat potential model is humidity, with a value of 82.444%. This correlates with Table 2, which shows the humidity variable to be a variable that is strongly associated with several other variables. Meanwhile, the parameter with the lowest relative contribution is the distance of the water source, with a value of 13.340%. Based on the graph in Figure 8b, the parameter or variable with the highest relative contribution to the Lowland Anoa habitat potential model is a human population with a

value of 88.891%. Meanwhile, the parameter with the lowest relative contribution is land use, with a value of 6.084%.

Figure 9 illustrates the histogram of each environmental parameter used for the Mountain Anoa (*Bubalus quarlesi*) training point extraction. It is known that the habitat characteristics of Mountain Anoa in terms of elevation are mostly in class 5 (>2500 m above sea level). When viewed from the land cover histogram, most are in class 10 (trees), indicating that Mountain Anoa prefer to live in forest areas. Regarding vegetation index, Mountain Anoa prefer areas with a high vegetation index, namely class 4 (0.5–0.75). Mountain Anoa also prefer areas that tend to be cold or in temperature class 1 (<20 °C). Water distance tends to be in areas close to water sources (rivers and lakes), in class 1 (0–500 m). Meanwhile, the humidity is in the range of 11–13%. From the disturbance factor, the farther away from the road or transportation the Mountain Anoa can live is directly proportional to the human population factor.

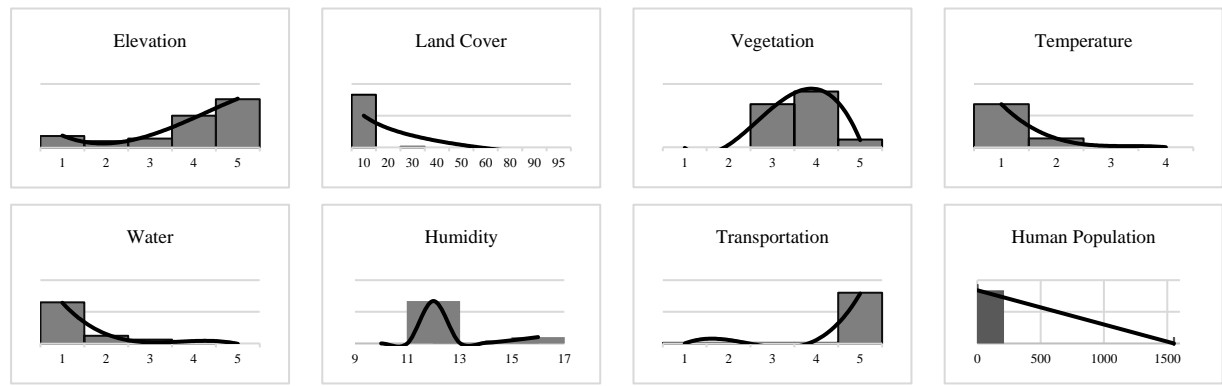

**Figure 9.** Histogram of training point parameters of Mountain Anoa (*Bubalus quarlesi*).

Figure 10 illustrates the histogram of each environmental parameter used to extract Lowland Anoa (*Bubalus depressicornis*) training points. It is known that the habitat characteristics of Lowland Anoa in terms of elevation are mostly in class 1 (0–1000 masl). When viewed from the land cover histogram, most are in class 10 (trees), indicating that Lowland Anoa prefer to live in forest areas. In terms of vegetation index, Lowland Anoa prefer areas with a very high vegetation index, namely class 6 (0.75–1). Lowland Anoa also prefer areas that tend to be moderate or in temperature class 2 (20–25 °C). Water distance tends to be in areas close to water sources (rivers and lakes), in class 1 (0–500 m). Meanwhile, the humidity is in the range of 17–18%. From the disturbance factor, the farther away from the road or transportation the Mountain Anoa can live is directly proportional to the human population factor.

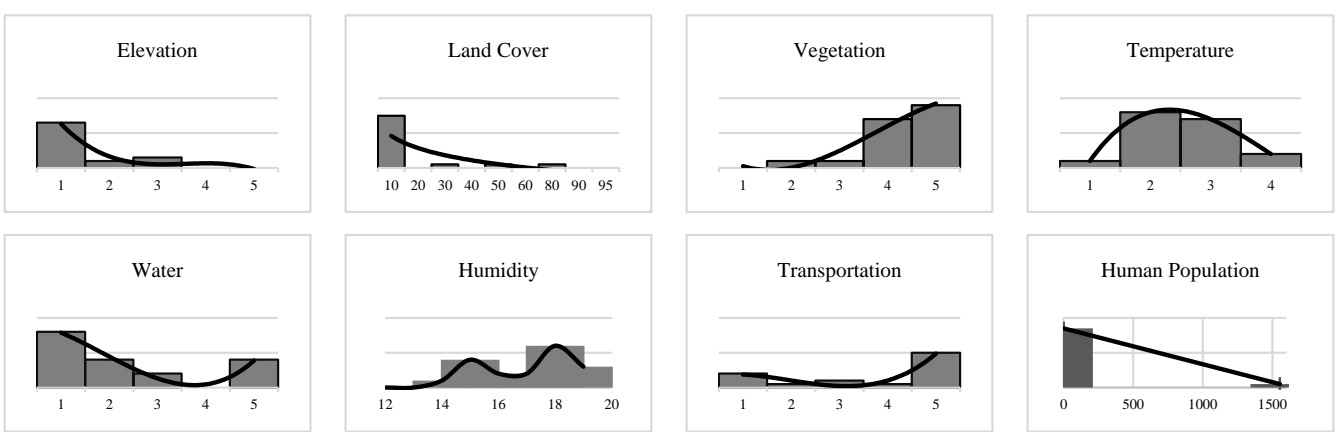

**Figure 10.** Histogram of training point parameters of Lowland Anoa (*Bubalus depressicornis*).

## 4. Discussion

The modeling of the habitat potential of Mountain Anoa (*Bubalus quarlesi*) and Lowland Anoa (*Bubalus depressicornis*) is supported using eight environmental variables, namely Elevation, Temperature, Vegetation, Land Cover, Transportation, Water Distance, Humidity and Human Population. The highest variable contribution for the Mountain Anoa habitat potential model is humidity with a value of 82.444% while the highest variable contribution result for the Lowland Anoa habitat potential model is human population with a value of 88.891%.

Comparisons of RF and MaxEnt methods have their own characteristics that depend on the research case. The RF method has been widely applied in the classification of remote sensing image data due to its insensitivity to noise and excessive training data and its good performance [28]. Prediction results from RF are obtained through the most results from each individual decision tree (voting for classification and averaging for regression). Because RF is the result of the most votes from each decision tree, it will issue more accurate predictions and results. This is because the more decision trees used to vote, the more accurate the resulting data will be. The principle of MaxEnt is to find the probability distribution of maximum entropy subject to a set of constraints derived from the species' occurrence data [11].

Based on this research, the RF method is better than the MaxEnt method. This is based on the AUC value, which is used as a general evaluation metric to measure the performance of classification models. AUC reflects the model's ability to distinguish between two different classes or categories [29]. AUC is theoretically and empirically proven to be better than accuracy metrics for evaluating classifier performance and distinguishing optimal solutions during classification training [30]. In the results of the *Bubalus quarlesi* habitat potential model, the RF results with a splitting ratio of training-testing 75:25 are the best results with an AUC value of 0.987. While in the results of the *Bubalus depressicornis* habitat potential model, the RF results with a splitting ratio training-testing 70:30 are the best results with an AUC value of 0.967. This is supported by research conducted by Zhao that the comparison of methods used in amphibian prediction in China resulted in the RF method being slightly better than the MaxEnt method. From Zhao's research, RF may be more applicable in predicting the native potential distribution of species with sufficient species occurrence data, given the additional predictive detail, the simplicity of use, the computational time involved and operational complexity. However, another study by Kaky [31] suggests that, if an area only has a presence data format, then in that situation MaxEnt is a better choice than a complex and computationally intensive "black box" ensemble. MaxEnt can promote practical conservation goals more effectively. This means that the accuracy of predictions is influenced by several environmental variables as well as the characteristics of the case study used, so not all modeling performed shows that RF is better than other methods.

## 5. Conclusions

The results of the comparison of MaxEnt and RF machine learning algorithms for the potential habitat showed that the RF 75:25 model is the best algorithm for modeling Mountain Anoa habitat potential with a high habitat potential level of 837,400 ha (5% of the island of Sulawesi), with AUC values of 0.987; acc 0.975; kappa 0.992; SN 0.944; SP 1; and DR 0.425. Meanwhile, the best model for Lowland Anoa habitat potential is the RF 70:30 algorithm with a high potential area of 6,015,700 ha (36% of the island of Sulawesi), for an AUC value of 0.967; ACC 0.909; kappa 0.814; SN 0.8; SP 1; and DR 0.364. The human population parameter has the highest relative contribution rate of 89% of the model. This indicates that Anoa extinction is very sensitive to the presence of a human population and is directly proportional to the lack of potential habitat. Information from the habitat modeling study of Mountain Anoa and Lowland Anoa can be used by IUCN in community or NGO collaboration efforts and advocacy for the conservation of vulnerable animals.

**Author Contributions:** Conceptualization, L.M.J. and D.A.; in-situ data, S.A.; methodology, L.M.J. and D.A.; formal analysis, L.M.J. and D.A.; writing—original draft preparation, D.A.; writing—review and editing, L.M.J., M.P.T., M.I. and A.A.W.; visualization, D.A.; supervision, L.M.J., M.P.T. and M.I. All authors have read and agreed to the published version of the manuscript.

**Funding:** This research was funded by Institut Teknologi Sepuluh Nopember, grant number 1334/PKS /ITS/2021.

**Data Availability Statement:** Shuttle Radar Topography Mission (DEM) data courtesy of the U.S. National Aeronautics and Space Administration; MODIS (LST and NDVI) data courtesy of the U.S. Geological Survey; ESA WorldCover; Human Population WorldPop, Relative Humidity FLDAS: Famine Early Warning Systems Network (FEWS NET) Land Data Assimilation System; Road, Lake, and Rivers (Waters) data courtesy of the Indonesia Information Geospatial Agency.

**Acknowledgments:** The author would like to thank the collaborative research team from University of Indonesia, University of West Sulawesi, IPB University and Insitut Teknologi Sepuluh Nopember who have helped provide facilities for supporting in-situ data in this research.

**Conflicts of Interest:** The authors declare no conflict of interest.

## Appendix A

**Table A1.** In-situ data on the coordinates of Anoa tracks.

| No | Spesies | Longitude (°) | Latitude (°) | Source |
|---|---|---|---|---|
| 1 | *Bubalus quarlesi* | 120.51870 | −1.57008 | [32] |
| 2 | *Bubalus quarlesi* | 120.04250 | −0.80722 | [33] |
| 3 | *Bubalus quarlesi* | 120.04050 | −0.81842 | [33] |
| 4 | *Bubalus quarlesi* | 120.04510 | −0.80831 | [33] |
| 5 | *Bubalus quarlesi* | 120.21340 | −1.52712 | [34] |
| 6 | *Bubalus quarlesi* | 120.00920 | −1.68975 | [34] |
| 7 | *Bubalus quarlesi* | 123.80960 | 0.47658 | [35] |
| 8 | *Bubalus quarlesi* | 123.80990 | 0.46652 | [35] |
| 9 | *Bubalus quarlesi* | 121.00570 | 0.60439 | [36] |
| 10 | *Bubalus quarlesi* | 120.00180 | −0.82303 | [37] |
| 11 | *Bubalus quarlesi* | 120.01500 | −0.80661 | [37] |
| 12 | *Bubalus quarlesi* | 120.03370 | −0.83094 | [37] |
| 13 | *Bubalus quarlesi* | 120.01270 | −0.82683 | [37] |
| 14 | *Bubalus quarlesi* | 120.77800 | −2.20443 | [38] |
| 15 | *Bubalus quarlesi* | 120.96800 | −1.69737 | [38] |
| 16 | *Bubalus quarlesi* | 120.97300 | −2.39240 | [38] |
| 17 | *Bubalus quarlesi* | 119.87400 | −3.07609 | [38] |
| 18 | *Bubalus quarlesi* | 119.39188 | −2.87850 | - |
| 19 | *Bubalus quarlesi* | 119.39369 | −2.87648 | Footprints |
| 20 | *Bubalus quarlesi* | 119.39366 | −2.87647 | Resting Place |
| 21 | *Bubalus quarlesi* | 119.39366 | −2.87646 | Footprints |
| 22 | *Bubalus quarlesi* | 119.39450 | −2.87313 | 2 Year Old Juvenile Female Stool—1 Week Stool Age—11 cm × 10 cm |
| 23 | *Bubalus quarlesi* | 119.38334 | −2.87160 | Traces—1 Month—3 Years Old |
| 24 | *Bubalus quarlesi* | 119.38450 | −2.86527 | Parent Footprints—1 Week—5.5 cm × 6.5 cm |
| 25 | *Bubalus quarlesi* | 119.38459 | −2.86526 | Child's Footprints—1 Week—3 cm × 3.5 cm |
| 26 | *Bubalus quarlesi* | 119.38456 | −2.85850 | Male Stool—1 Week Stool Age |
| 27 | *Bubalus quarlesi* | 119.38484 | −2.84280 | 3 Day Trail 6 × 7—1.5 cm deep |
| 28 | *Bubalus quarlesi* | 119.38478 | −2.84275 | 3 Month Old Stool—13.5 × 14 |
| 29 | *Bubalus quarlesi* | 119.38467 | −2.84297 | 8 Month Anoa Trail—4.5 × 5.5 |
| 30 | *Bubalus quarlesi* | 119.38460 | −2.84316 | Male Stool—1 Week Old—20.5 × 3 |
| 31 | *Bubalus quarlesi* | 119.38274 | −2.84330 | Anoa Trail 1 Week—6 × 8. 1.4 cm deep |
| 32 | *Bubalus quarlesi* | 119.38276 | −2.84331 | 3-Month Male Stool—26 × 13.5 |
| 33 | *Bubalus quarlesi* | 119.38274 | −2.84322 | Nest 82 cm × 41 cm |
| 34 | *Bubalus quarlesi* | 119.38277 | −2.84351 | - |
| 35 | *Bubalus quarlesi* | 119.38277 | −2.84351 | Male Trail—7 Years Old—3 Day Trail—7.4 cm × 7 cm—2 cm Depth |
| 36 | *Bubalus quarlesi* | 119.38276 | −2.84351 | 7 Year Old Female Trail—3 Day Trail—4 cm × 7 cm—1.5 cm Depth |
| 37 | *Bubalus quarlesi* | 119.38561 | −2.84361 | 5 Day Trail—6 × 8—2 cm deep |
| 38 | *Bubalus quarlesi* | 119.38596 | −2.84350 | 2 Day Trail—5.3 cm × 8 cm—2 cm Depth |
| 39 | *Bubalus quarlesi* | 119.38623 | −2.84363 | Resting Place—90 cm × 80 cm × 40 cm |
| 40 | *Bubalus quarlesi* | 119.38655 | −2.84368 | Juvenile Stool < 1 Year—1 Week Stool Age—6.5 cm × 7.5 cm |
| 41 | *Bubalus quarlesi* | 119.38652 | −2.84372 | Child Trail (Youth)—1 Week Trail Age—4 cm × 5.7 cm |

**Table A1.** *Cont.*

| No | Spesies | Longitude (°) | Latitude (°) | Source |
|---|---|---|---|---|
| 42 | *Bubalus quarlesi* | 119.38697 | −2.84411 | Footprints of 5 Year Old Male—Age of Footprints 2 Days—6 cm × 7 cm—Depth 0.9 cm—Blunt Hooves |
| 43 | *Bubalus quarlesi* | 119.38616 | −2.84435 | Male Stool—1 Month Stool Age—9.5 cm × 10 cm |
| 44 | *Bubalus quarlesi* | 119.38577 | −2.84400 | Female Trail 4–5 Years—Trail Age 8 Days—6.5 cm × 7.5 cm—Depth 1 cm |
| 45 | *Bubalus quarlesi* | 119.38574 | −2.84397 | 4–5 Year Old Female Manure—2 Weeks Manure Age—20.5 cm × 23.5 cm |
| 46 | *Bubalus quarlesi* | 119.38575 | −2.84386 | 2.5 Year Old Male Trail—3 Day Trail Age—6.6 cm × 4.5 cm—2 cm Depth |
| 47 | *Bubalus quarlesi* | 119.38567 | −2.84387 | Female Resting Place—90 cm × 60 cm × 35 cm |
| 48 | *Bubalus quarlesi* | 119.38565 | −2.84387 | Female Trail—2 Weeks of Trail Age—6 cm × 7 cm—1 cm Depth |
| 49 | *Bubalus quarlesi* | 119.38482 | −2.84401 | 1.5 Year Old Male Imprint—3 Weeks Manure Age—10.5 cm × 10 cm |
| 50 | *Bubalus quarlesi* | 119.38482 | −2.84386 | 1.5 Year Old Male Trail—3 Week Trail—5 cm × 6.5 cm—2 cm Depth |
| 51 | *Bubalus quarlesi* | 119.38323 | −2.84537 | Footprint—6 cm × 7.5 cm—2.3 cm depth |
| 52 | *Bubalus quarlesi* | 119.38309 | −2.84509 | Footprint—5.3 cm × 8 cm—3 cm depth |
| 53 | *Bubalus quarlesi* | 119.38311 | −2.84484 | Stools—16 cm × 10 cm |
| 54 | *Bubalus quarlesi* | 119.38872 | −2.84201 | 4 Day Female Trail—5 cm × 7 cm—1.4 Depth |
| 55 | *Bubalus quarlesi* | 119.38878 | −2.84194 | 4 Day Male Trail—7 cm × 6 cm—1.5 cm depth |
| 56 | *Bubalus quarlesi* | 119.39028 | −2.83982 | Lantalomo Peak |
| 57 | *Bubalus quarlesi* | 119.39047 | −2.83873 | 3 Month Male Manure—9 cm × 12 cm |
| 58 | *Bubalus quarlesi* | 119.39124 | −2.83710 | 3 Day Female Trail—6 cm × 6 cm—1 cm depth |
| 59 | *Bubalus quarlesi* | 119.39193 | −2.83586 | 1 Month Female Manure—14 cm × 12 cm |
| 60 | *Bubalus quarlesi* | 119.39194 | −2.83584 | 1.5 Year Old Child Imprint—2 Weeks Imprint Age—3 cm × 5 cm—Depth |
| 61 | *Bubalus quarlesi* | 119.39316 | −2.83215 | 2 Day Female Trail—6 cm × 8 cm—3 cm depth |
| 62 | *Bubalus quarlesi* | 119.39316 | −2.83197 | Female footprint 2–3 yrs—1 week footprint—6 cm × 8 cm—2 cm deep |
| 63 | *Bubalus quarlesi* | 119.39289 | −2.82419 | 2 Day Female Trail—6 cm × 6 cm—2 cm depth |
| 64 | *Bubalus quarlesi* | 119.38518 | −2.82199 | The Nest |
| 65 | *Bubalus quarlesi* | 119.38431 | −2.82164 | Stools—18 cm × 20 cm |
| 66 | *Bubalus quarlesi* | 119.38346 | −2.81992 | Stools—14 cm × 15 cm |
| 67 | *Bubalus quarlesi* | 119.38342 | −2.81958 | Stools—16 cm × 13 cm |
| 68 | *Bubalus quarlesi* | 119.38349 | −2.81960 | Male Manure 1 Day—9 cm × 8 cm |
| 69 | *Bubalus quarlesi* | 119.38172 | −2.81800 | Female Manure < 1 Year Old—1 Week Manure—12 cm × 14 cm |
| 70 | *Bubalus quarlesi* | 119.38175 | −2.81802 | 2 Month Old imprint—8 cm × 9 cm—3 cm depth |
| 71 | *Bubalus quarlesi* | 119.38089 | −2.81808 | Male Tracks Age > 1 Year—3 Day Tracks—5 cm × 6.5 cm—2 cm depth |
| 72 | *Bubalus quarlesi* | 119.38098 | −2.81804 | 5–6 Year Old Female Stool—5 Day Stool—29.5 cm × 18 cm |
| 73 | *Bubalus quarlesi* | 119.38108 | −2.81791 | 1 Day Female Trail—6.3 cm × 7 cm—0.6 cm Depth |
| 74 | *Bubalus quarlesi* | 119.38101 | −2.81792 | 1 Day Male Footprint—4.5 cm x 5 cm—1 cm depth |
| 75 | *Bubalus quarlesi* | 119.37827 | −2.81687 | 4 Day Female Trail—4.5 cm × 8 cm—1 cm depth |
| 76 | *Bubalus quarlesi* | 119.37728 | −2.81484 | Nest—95 cm × 113 cm × 64 cm |
| 77 | *Bubalus quarlesi* | 119.37725 | −2.81487 | Resting Place |
| 78 | *Bubalus quarlesi* | 119.37707 | −2.81525 | 1 Day Female Trail—6 cm × 9 cm—1 cm depth |
| 79 | *Bubalus quarlesi* | 119.37729 | −2.81541 | Male Footprints 1 Day—5.5 cm x 6.3 cm—0.5 cm Depth |
| 80 | *Bubalus quarlesi* | 119.37739 | −2.81538 | 1 Week Female Manure—15.5 cm × 14 cm |
| 81 | *Bubalus quarlesi* | 119.37747 | −2.81556 | 2 Year Old Male Stool—1 Day Stool Age—13.7 cm × 9 cm |
| 82 | *Bubalus quarlesi* | 119.37746 | −2.81560 | Female Trail 4–5 years old—Trail Age 1 Day—5.1 cm × 7 cm—Depth 3 cm |
| 83 | *Bubalus quarlesi* | 119.38848 | −2.82364 | 1.5 Year Male Stool—1 Day Stool Age—10 cm × 8 cm |
| 84 | *Bubalus quarlesi* | 119.39298 | −2.82827 | Parent male 1.5 years old—1 day old—4.5 cm × 6 cm—0.3 cm depth |
| 85 | *Bubalus depressicornis* | 122.12180 | −4.49525 | [39] |
| 86 | *Bubalus depressicornis* | 120.52510 | −1.58031 | [32] |
| 87 | *Bubalus depressicornis* | 120.52360 | −1.57728 | [32] |
| 88 | *Bubalus depressicornis* | 120.51810 | −1.56668 | [32] |
| 89 | *Bubalus depressicornis* | 123.76800 | 0.51531 | [38] |
| 90 | *Bubalus depressicornis* | 122.61000 | 0.62516 | [38] |
| 91 | *Bubalus depressicornis* | 120.21500 | −1.55003 | [38] |
| 92 | *Bubalus depressicornis* | 120.79300 | 0.66886 | [38] |
| 93 | *Bubalus depressicornis* | 119.61900 | −1.30254 | [38] |
| 94 | *Bubalus depressicornis* | 122.06100 | −1.13299 | [38] |
| 95 | *Bubalus depressicornis* | 121.87800 | −4.45523 | [38] |
| 96 | *Bubalus depressicornis* | 122.80600 | −4.20800 | [38] |
| 97 | *Bubalus depressicornis* | 122.87000 | −4.35533 | [38] |
| 98 | *Bubalus depressicornis* | 122.72800 | −4.47538 | [38] |
| 99 | *Bubalus depressicornis* | 119.43333 | −5.15000 | [40] |
| 100 | *Bubalus depressicornis* | 120.02000 | −4.27083 | [40] |
| 101 | *Bubalus depressicornis* | 121.08086 | −2.00000 | [40] |
| 102 | *Bubalus depressicornis* | 122.23148 | −4.11590 | [40] |

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
