# Peer review of "Spatial Analysis of Mountain and Lowland Anoa Habitat Potential Using the Maximum Entropy and Random Forest Algorithm"

_world, doi:10.3390/world4040041_

Round 1

Reviewer 1 Report

Dear Authors, Dear Editors,

Thank you for giving me the opportunity to review the manuscript on the ”Spatial Analysis of Mountain and Lowland Anoa Habitat Potential using The Maximum Entropy and Random Forest Algorithm”.

The manuscript brings information on the distribution and habitat suitability of the two Anoa species on the island of Sulawesi, being important for the knowledge of these species’ ecology, and therefore, it might have conservation implications, given their endangered status. It has also methodological value, through the comparison of the results obtained by using the two methods for species distribution modeling.

The study is based on a good dataset and the manuscript is generally easy to follow, but is not suitable for publishing in this form. My main comment concerns the lack of proper discussions (the Discussion section comprises actually mostly results and has absolutely no citation) and the poor organization of the manuscript (methods mixed with results, results mixed with discussions, conclusions which are mainly a repetition of the abstract).

The abstract also needs rewriting. It has too many numbers. For instance, instead of focusing on the exact surface expressed in ha, I would suggest leaving in the abstract only the percentage values, which are more palatable and meaningful for the general readership. Also, instead of giving all the exact parameters of the best model (these details are ok in the results section, but not in the abstract), I would suggest describing it in words – for instance saying what factors are the most important and which are the implications. Giving so many acronyms without any explanation is also not ok.

Other minor comments are in the attached PDF.

English language is mainly ok, I have just some minor observations and corrections, mentioned in the text. Be consistent with the use of capital letters and always write the Latin names of species (and subspecies) in italic font.

Author Response

Dear Reviewer 1

We have response your comment and suggestion on attachment below. Please also check our revised manuscript.

Thank you very much for your great effort in improving the quality of our manuscript.

Best regards

Reviewer 2 Report

Overall, this is a very good paper, and I was very glad to read and review it. 

In the first place, it is extremely well written.  The flow of information is well organized.  And the data presentations are succinct and complete.  In these respects, this paper could serve as an example of how to present a complex finding in a clear and concise fashion. 

Even more significantly, the subject matter (Anoa distribution and conservation) is both important and timely.  This line of investigation holds forth the prospect of providing essential information to local conservation teams. 

However, having said all of that, I do think that there are some issues with the paper that need to be addressed.  Some are substantive; others are merely technical. 

Substantive Issues

The data collected on Anoa distribution is inadequately described.  Indeed, the only information is provided in lines 83-84 as “The data required for this study included in-situ data on the coordinates of anoa 83 tracks from several sources.”  This is simply not acceptable, and it is a major flaw of this paper.  The method section of any scientific report should be sufficiently complete to allow the study to be replicated in exact detail.  Moreover, the very validity and applicability of this paper depends upon the representativeness of the anoa-distribution data set.  For both reasons, two things are needed.  Some justification for the use of footprints as representative of distribution.  And much more information must be provided on where, how, and when the footprint data was collected.  This is essential. 

In the Introduction (line 67) and in the Conclusions (line 268), a real opportunity is lost in describing the potential benefit of this study.  The Introduction could/should say more about why the forth-coming data will be useful.  And the authors should provide recommendations in the Conclusions section pertaining to Anoa conservation that can be derived from their findings. 

Similarly, the authors missed an opportunity to expand on the implications that the Random Forest algorithm provided a better fit than the Maximum Entropy algorithm.  Would they recommend the RF model over the ME model to other researches?  Why or why not?

Technical Issues

The use of bracketed numbers (e.g., [5]) is acceptable/appropriate when attached to names or statements, but they cannot be used in place of names.  Thus, the following examples are acceptable:  The Isle of Mann lies close to England [6].  OR  Smith and Jones [7] list the Amazon River as essential habitat for the Pink Dolphin.  Whereas, the following is not acceptable:  [7] lists the Amazon River as essential ……  Mistakes of this kind were noticed on lines 45, 52, and 65. 

On line 96, the term google earth should be capitalized, as it already is elsewhere in this manuscript.

Throughout the paper, the use of the word “population” is at times unclear.  With effort, I was able to deduce that it referred to human population values, not anoa population values.  In my opinion it would help the reader by clarifying this by changing “population” to “human population” (some examples:  lines 24, 90, 278). 

I emphasize, however, that all of these recommendations are offered on the hope that they will help to improve an already-good paper.  The bottom line is that these authors should be congratulated for bringing this excellent project to fruition. 

Author Response

Dear Reviewer 2

We have responded to your comment and suggestion in the attachment below. Please also check our revised manuscript.

Thank you very much for your great effort in improving the quality of our manuscript.

Best regards

Reviewer 3 Report

The manuscript is a significant contribution in the field of habitat suitability modeling of Mountain and Lowland Anoa. Authors have used the best possible combination of specific variables via two algorithms. I have some suggestions for the improvement of the manuscript in the attached file.

Miner editing required

Author Response

Dear Reviewer 3

We have responded to your comment and suggestion in the attachment below. Please also check our revised manuscript.

Thank you very much for your great effort in improving the quality of our manuscript.

Best regards

Round 2

Reviewer 1 Report

Dear Authors,

I’ve read the revised version of the manuscript on the ”Spatial Analysis of Mountain and Lowland Anoa Habitat Potential using The Maximum Entropy and Random Forest Algorithm”.

Although the revised version is much improved, and most of my previous comments and suggestions were properly addressed, there are still some issues that need to be solved before I can full heartedly recommend the manuscript for publication.

First, although the manuscript was in part re-organized and some parts correctly separated, there is still some mixing between Methods and Results (this section includes some analysis description and details, with the corresponding citations, that needs to go into Methods).

Second, in the newly introduced Table 3, you show that temperature, humidity and elevation are strongly correlated. Therefore it is not correct to enter the analysis all of them, you should keep just one (the one that is more biologically important or the one that is more easy to interpret), else the results may be biased. Check the literature on this matter and make the needed changes.

Other minor observations are in the text from the attached PDF.

English language is fine, the are only some minor corrections to be made - they are mentioned in the PDF.

Author Response

Dear Reviewer

We add some modification based on your suggestion on two separated files: replay to reviewer and revised manuscript.

Thanks 

Reviewer 3 Report

The manuscript has been now in a publishable form.

Miner language editing required

Author Response

Dear Reviewer

Minor correction based on your suggestion has been made on the revised manuscript.

Thanks for your great suggestion

Round 3

Reviewer 1 Report

Dear Authors,

I have read the second version of your revised manuscript on Anoa habitat modeling and I consider that you have properlya ddressed my comments and suggestions, so I recommend the publication of your manuscript. 

But you should return to the previous version of the title, including Maximum Entropy and Random forests instead the acronyms. I suggested you added the acronym (in parantheses) after the first mentioning of the two methods, (in the abstract and in the main text - introduction, so that the reader make the connection between the acronyms and their meaning), not to replace the full names everywhere.
